

# Development of machine learning models for the prediction of the skin sensitization potential of cosmetic compounds

Wu Qiao, Tong Xie, Jing Lu and Tinghan Jia

Pigeon Manufacturing (Shanghai) Co., Ltd., Shanghai, China

## ABSTRACT

**Background:** To enhance the accuracy of allergen detection in cosmetic compounds, we developed a co-culture system that combines HaCaT keratinocytes (transfected with a luciferase plasmid driven by the AKR1C2 promoter) and THP-1 cells for machine learning applications.

**Methods:** Following chemical exposure, cell cytotoxicity was assessed using CCK-8 to determine appropriate stimulation concentrations. RNA-Seq was subsequently employed to analyze THP-1 cells, followed by differential expression gene (DEG) analysis and weighted gene co-expression net-work analysis (WGCNA). Using two data preprocessing methods and three feature extraction techniques, we constructed and validated models with eight machine learning algorithms.

**Results:** Our results demonstrated the effectiveness of this integrated approach. The best performing models were random forest (RF) and voom-based diagonal quadratic discriminant analysis (voomDQDA), both achieving 100% accuracy. Support vector machine (SVM) and voom based nearest shrunken centroids (voomNSC) showed excellent performance with 96.7% test accuracy, followed by voom-based diagonal linear discriminant analysis (voomDLDA) at 95.2%. Nearest shrunken centroids (NSC), Poisson linear discriminant analysis (PLDA) and negative binomial linear discriminant analysis (NBLDA) achieved 90.5% and 90.2% accuracy, respectively. K-nearest neighbors (KNN) showed the lowest accuracy at 85.7%.

**Conclusion:** This study highlights the potential of integrating co-culture systems, RNA-Seq, and machine learning to develop more accurate and comprehensive *in vitro* methods for skin sensitization testing. Our findings contribute to the advancement of cosmetic safety assessments, potentially reducing the reliance on animal testing.

Corresponding author
Tinghan Jia, adolf@pigeon.cn

## INTRODUCTION

An important endpoint for consumer and occupational safety assessment is chemical-induced skin sensitization, which is an allergic reaction caused by repeated skin exposure to a single substance or mixture (*United Nations, 2017*). Continuous skin sensitization can lead to allergic contact dermatitis (ACD), a severe skin condition that affects consumer health and the experience of using cosmetic products (*Kimber et al.,*

*2002*). Following the implementation of Cosmetics Regulation 1223/2009 and similar legislation in 2009, traditional animal testing for cosmetic products and ingredients has been progressively banned. As a result, there has been a growing demand for non-animal testing methods, with increasing requirements for accuracy (*European Union, 2009*). Based on the complexity of the biological processes involved in allergic reactions, the OECD Test Guidelines break down the allergy Adverse Outcome Pathway (AOP) into four key events (KEs): (i) covalent binding of electrophilic substances to nucleophilic sites in skin proteins initiates downstream activation (*Enoch et al., 2011*); (ii) keratinocyte activation (*Emter, Ellis & Natsch, 2010*); (iii) dendritic cell (DC) activation (*Ashikaga et al., 2010*); (iv) proliferation of T cells in the lymph nodes (*Gerberick et al., 2007*). From key events (i) to (iii), various methods such as direct peptide reactivity assay (DPRA), KeratinoSens™ and the human cell line activation test (h-CLAT) have been developed, but their accuracy remains unsatisfactory. To improve the accuracy, the following integrated approaches to testing and assessment (IATA) methods have been established (*Rovida et al., 2015*).

The Genomic Allergen Rapid Detection (GARD) method employs transcriptome wide microarray analysis in a myeloid cell line to assess 200 transcriptomic biomarkers, which function as predictive signatures (*Johansson et al., 2011*, *2014*). Microarrays, which provide limited gene-related data, use pre-designed probes to capture specific mRNA sequences (*Karakach et al., 2010*). With the advancement of next-generation sequencing (NGS) technology, RNA-Seq has been widely applied to protein-coding genes, non-coding RNA, and immune genes, with the cost and sequencing time rapidly decreasing (*Qiao et al., 2024*). The researchers used RNA-Seq to analyze the changes in the expression of the THP-1 gene and molecular mechanism under different treatment conditions (*Zhang et al., 2016*). Weighted gene co-expression network analysis (WGCNA), which can cluster genes with similar expression patterns and examine the relationships between these modules, is widely applied in studies investigating the association between phenotypic traits and gene expression (*Langfelder & Horvath, 2008*).

Machine learning (ML), a branch of artificial intelligence, comprises numerous algorithms such as support vector machine (SVM) (*Cortes & Vapnik, 1995*), random forest (RF) (*Breiman, 2001*), K-nearest neighbors (KNN) (*Begum, Chakraborty & Sarkar, 2015*) and linear discriminant analysis (LDA) (*Goksuluk et al., 2019*). With the increase in data and advances in computing power, ML has found extensive applications in RNA-Seq data analysis, including gene expression data classification (*Wang et al., 2018*), feature selection (*Li et al., 2016*), differential gene expression analysis, and the development of classification and prediction models (*Johansson et al., 2014*). The GARD method achieves high sensitivity and accuracy by using SVM to classify and predict tested compounds. It has been established as one of the OECD 442E standard methods (*OECD, 2024*). Co-culture systems offer several advantages over monoculture-based assays, including facilitating intercellular crosstalk and enabling the combination of multiple key events in a single assay (*Thélu, Catoire & Kerdine-Römer, 2020*). To date, numerous THP-1 co-culture methods have been employed, primarily involving the co-culture of different cell types and the co-culture of reconstructed human epidermis. Hennen developed a co-culture model comprising HaCaT keratinocytes and THP-1 cells, which has demonstrated excellent
potential for identifying skin sensitizers (*Hennen & Blömeke, 2017*). *Galbiati et al. (2020)* showed that keratinocytes promote DC activation by co-culturing the epithelial-like cell line NCTC2544 with THP-1 cells. *Schellenberger et al. (2019)* combined RHE in coculture with THP-1 cells placed underneath RHE, applying compounds using topical exposure to avoid issues related to water solubility.

In this present study, we focused on the evaluation of RNA-Seq analysis of THP-1 in co-cultured system to predict sensitizing and non-sensitizing compounds. The co-cultured combines two different cell types (HaCaT transfected with a luciferase plasmid driven by the AKR1C2 promoter and THP-1) that address KE2 and KE3 of AOP. The overall experimental design, as depicted in Fig. 1. In the first step, we co-cultured HaCaT and THP-1 cells; compounds were added, and cytotoxicity was assessed. In a second step, we collected the THP-1 cells, extracted the RNA and performed transcriptome sequencing. Finally, data processing, machine learning model construction and prediction were performed. Based on the results of this study, we established a robust experimental design that compared different data processing methods, feature selection, and various machine learning models to evaluate their impact on prediction accuracy. These results provide valuable insights for future applications.

## MATERIALS AND METHODS

### Regents

A total of 15 (excluding blank) chemical compounds (Table 1), including 10 skin sensitizing compounds and 5 none-sensitizing compounds, were used for subsequent cells stimulation, RNA-Seq data generation, and supervised machine learning model construction. Tert-butylhydroquinone (THBQ), 2,4-dinitrochlorobenzene (DNCB), 2-mercaptobenzothiazole, phenylacetaldehyde, sodium dodecyl sulfate (SDS), isopropanol (IPA) and DMSO were purchased from Sigma-Aldrich (St. Louis, MO, USA). ρ-phenylenediamine, resorcinol, diphenylcyclopropenone, eugenol, isoeugenol, 1,2-dibromo-2,4-dicyanobutane, glycerol and squalane were purchased from Shanghai Macklin Biochemical Technology Co., Ltd., (Macklin, Shanghai, China).

### Cell culture and maintenance

THP-1, a human monocytic leukemia cell line, was purchased from Shanghai Zhong Qiao Xin Zhou Biotechnology Co., Ltd (Shanghai, China) and cultured in RPMI-1640 medium (Shanghai BasalMedia Technologies Co., LTD, Shanghai, China) with 10% fetal bovine serum (FBS) (Gibco, Waltham, MA, USA) and 50 µM 2-mercaptoethanol (Sigma Aldrich, St. Louis, MO, USA) at 37 °C and 5% $CO_2$. According to the OECD442E standards, THP-1 cells were routinely seeded every 2–3 days at a density of 0.1 to $0.2 \times 10^6$ cells/mL and maintained at densities of 0.1 to $1.0 \times 10^6$ cells/mL.

HaCaT (Human Keratinocytes Cells), harboring a luciferase reporter gene plasmid under the control of AKR1C2 gene promoter, was engineered in-house. HaCaT cells were cultured in DMEM medium (Cytiva, Burlington, MA, USA) + 10%FBS (Gibco, Waltham, MA, USA) + 200 µg/mL G418 (Sigma Aldrich, St. Louis, MO, USA) to maintain the gene

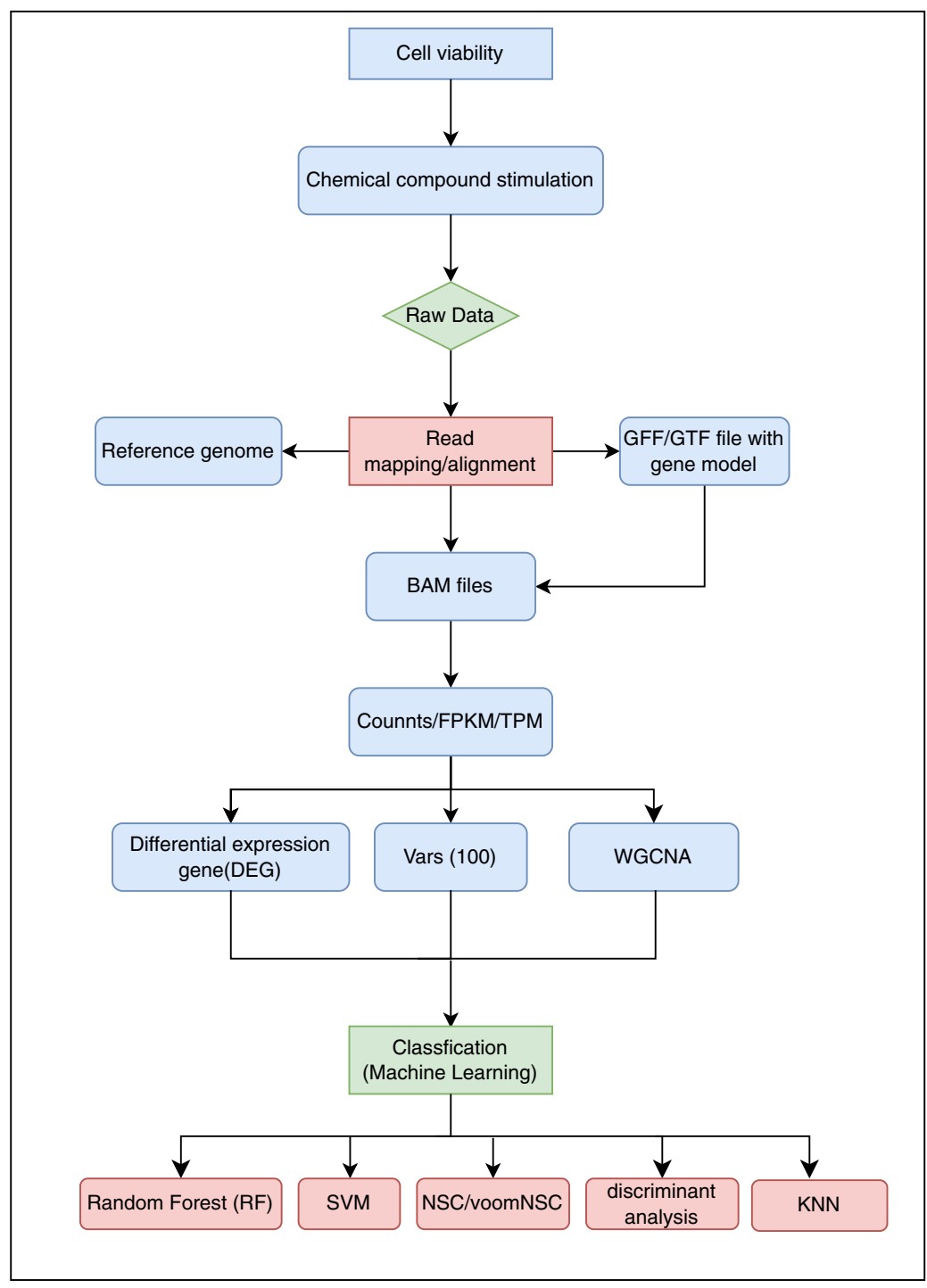

**Figure 1 Flow chart of the experiment.** GFF, general feature format; GTF, gene transfer format; FPKM, fragment per kilobase of transcript; TPM, transcripts per million; WGCNA, weighted gene co-expression network analysis; SVM, support vector machine; NSC, nearest shrunken centroids; KNN, K-nearest neighbor.

**Table 1 Chemical compounds.**

| Compound | Abbreviation | Chemical abstract service (CAS) | Pro-/pre-hapten |
|---|---|---|---|
| Tert-butylhydroquinone | THBQ | 1948-33-0 | Prehapten |
| 2,4-dinitrochlorobenzene | DNCB | 97-00-7 | Hapten |
| ρ-phenylenediamine | Phenyle | 106-50-3 | Prehapten |
| Resorcinol | Resorcinol | 108-46-3 | Prohapten |
| 2-mercaptobenzothiazole | Mercap | 149-30-4 | Hapten |
| Diphenylcyclopropenone | Dipheny | 886-38-4 | Hapten |
| Eugenol | Eugenol | 97-53-0 | Prohapten |
| Phenylacetaldehyde | Phenylac | 122-78-1 | Hapten |
| Isoeugenol | Isoeugenol | 97-54-1 | Prehapten |
| 1,2-dibromo-2,4-dicyanobutane | bromoml | 35691-65-7 | Hapten |
| Sodium dodecyl sulfate | SDS | 151-21-3 | – |
| Isopropanol | IPA | 67-63-0 | – |
| Glycerol | Glycerol | 56-81-5 | – |
| Squalane | Squalane | 111-01-3 | – |
| Dimethyl sulfoxide | DMSO | 67-68-5 | – |
| Blank | Blank | Blank | – |

at 37 °C and 5% $CO_2$ in a humidified atmosphere. Cells should avoid grow to full confluence.

## Co-culture and chemical compound treatment

The co-culture system was set up according to the following described method. In brief, when the HaCaT cells reached 80–90% confluency, they were digested with 0.25% trypsin (Cytiva, Burlington, MA, USA), collected, and seeded into 96-well plates at a density of 10,000 cells per well for 24 h. THP-1 cells were collected, centrifuged, and re-suspended in fresh RPMI-1640 medium supplemented with 10% FBS at $2 \times 10^6$ cells/mL. Then cells are distributed into HaCaT-seeded 96 wells plate with 80 μL ($1.6 \times 10^5$ cells/well).

The compounds listed in Table 1 were dissolved in DMSO to create stock solutions with a concentration of 500 mg/mL. Eleven concentrations were subsequently prepared using two-fold serial dilutions. These stock solutions were then further diluted 250-fold in RPMI-1640 + 10% FBS medium. Finally, 80 μL of the prepared solutions were added 1:1 to the treated 96-well plates and incubated for 48 h at 37 °C and 5% $CO_2$.

## Cell viability

After 48 h of treatment, the treated cells were collected 100 μL cells per well. The supernatant medium was added 10 μL of CCK-8 (Life-iLab, Shanghai, China) solution to each well and incubated the plate in the incubator for 2–4 h. THP-1 cell viability was determined using standard calculation methods.

## RNA-Seq

Bulk RNA from THP-1 cells was extracted using TRIzol (Thermo Fisher Scientific, Waltham, MA, USA) user guide. The concentration and purity of RNA samples were evaluated using Nanodrop 2000 spectrophotometer (Thermo Fisher Scientific, Waltham, MA, USA) and checked on Agilent 2100 Bioanalyzer. The mRNA was enriched by TIAN-Seq mRNA Capture (TIANGEN Biotech, Beijing, China), the transcriptome sequencing library was construct using TIANSeq Fast RNA Library kit. After amplification using the cBot cluster generation system, RNA libraries were sequenced using the Illumina platform NovaSeq 6000 to obtain 150 bp paired end reads. RNA-Seq data had been deposited in the NCBI Sequence Read Archive (SRA) under BioProject accession number PRJNA1148804.

## Bioinformatics

### General

Clean data of FASTQ format were obtained by removing adapters and low-quality base with Trim_galore (v0.6.10). Paired-end clean reads were aligned to *Homo sapiens* reference genome (GRCh38) using Hisat2 (v2.2.1), and Rsubread (v2.12.3) was used to count the reads number mapped to each gene. The obtained raw reads data were used for subsequent analysis. Image visualization was performed using ggplot2 (v3.5.1). The RNA-Seq data processing was conducted using a workflow constructed with Snakemake. For detailed usage instructions and workflow configuration, please refer to the GitHub repository available at https://github.com/erwinQiao/rna-seq-hisat2-featureCounts.

### Principal component analysis

For PCA analysis, DESeqDataSet values from varianceStabilizingTransformation (vst) were adjusted using the removeBatchEffect function in limma (3.54.2) to remove batch effects. As shown in Fig. S1, vst-transformed data were clustered using the hclust function and outlier samples were excluded before proceeding with subsequent analyses. PCA plot was generated using plotPCA function from the DESEq2 packages.

### Differential expression gene analysis

Differential expression gene were generated using the DESeq function from DESeq2 package using the default setting (*Love, Huber & Anders, 2014*). Samples were grouped according to sensitizing properties of the compounds (Positive or Negative). The negative group was used as the reference standard for differential analysis and Significant RNA expression was defined as those with adjust $p$-value < 0.05 and $|\log_2 FC| > 1$. A volcano plot was generated using the ggplot2 package, and a heatmap was created using the pheatmap package. Differential expression genes were selected based on the criteria of $|\log_2 FoldChange| > 1.5$ and adjust $p$-value < 0.05.

### Kyoto Encyclopedia of Genes and Genomes and Gene Ontology

Enrichment analyses of KEGG pathways and GO terms were performed using enrichKEGG and enrichGO functions from the clusterProfiler package in R (*Yu et al., 2012*). Differentially expression gene sets were using log2FoldChange and adjusted $p$-value.
A multiple correction has been performed with the *p*-adjust function using the "BH" method. Both KEGG and GO visualizations were created using the ggplot2 package.

### WGCNA

We utilized WGCNA package in R to construct a co-expression network following the general steps outlined below: 1. The pickSoftThreshold function was employed to identify the optimal soft thresholding power (β). 2. bolckwiseModules function was used to con-struct the automatic network and module detection. 3. To relate modules to allergic traits, gene significance (GS) and module membership (MM) were calculated. The corresponding gene information of the modules was extracted for further analysis. Genes were filtered based on their weight, and the selected network was visualized using Cytoscape software (v3.10.1).

### Machine-learning algorithms for classification

We used the MLSeq package, which is a comprehensive tool for applying machine learning algorithms to the classification of RNA-Seq data. MLSeq input data consisted of raw counts, preprocessing included transformation and normalization. We compared two preprocessing methods: deseq-vst and trimmed mean of M means (tmm)-logcpm. Additionally, we assessed the impact of three feature selection approaches on the results: (1) the top 100 features with the highest gene-wise variances; (2) differential expression analysis to select 50 upregulated genes and 50 downregulated genes; and (3) selecting the top 100 genes with the highest weights from allergen-associated gene modules identified by WGCNA.

We split the data into two parts: 60% for the training dataset and 40% for the test dataset. In this experiment, the tenfold cross-validation technique was used to calculate the model performance based on the scores. In this study, we employed eight different machine learning methods, including support vector machine (SVM), random forest (RF), k-nearest neighbors (KNN), nearest shrunken centroids (NSC)/voom nearest shrunken centroids (voomNSC), poison linear discriminant analysis (PLDA) and negative binomial linear discriminant analysis (NBLDA), voom-based diagonal quadratic discriminant analysis (voomDQDA), and voom-based diagonal linear discriminant analysis (voomDLDA). All algorithms (Fig. 1) were used with default parameters.

## Statistical analysis

Cell assays were performed in triplicate in this study and data are presented as mean ± standard deviation (SD). RNA-Seq data were subjected to multiple testing correction using the Benjamini-Hochberg method. An adjusted *p*-value and *p*-value < 0.05 were considered significant. The allergic group and non-allergic group contained 18 and six samples respectively, with a minimum of two technical replicates. The statistical power of this experimental design, calculated using RNASeqPower (1.38.10), is 0.803. The results of cell viability were graphed using GraphPad Prism 9, while other results were visualized using R 4.0.3.

## RESULTS

### Selection of chemical compounds concentrations

The cytotoxicity of 15 test compounds listed in Table 1 was assessed using THP-1 cells across a concentration range of 1,000 to 3.9 μg/mL. Cell viability was assessed after 48 h of treatment using the CCK-8 assay.

Among the 15 chemical compounds, IPA, glycerol, squalane and DMSO exhibited the lowest cytotoxicity, with IC50 values greater than 1,000 μg/mL. To identify suitable concentrations for subsequent experiments, we determined the highest concentration for each compound that maintained at least 80% cell viability. Based on the results shown in Fig. 2, the final concentrations of the samples used were determined as follows: double THBQ concentration (dTHBQ): 62.5 μg/mL; THBQ: 31.25 μg/mL; double DNCB (dDNCB): 3.9 μg/mL; DNCB: 1.95 μg/mL; double ρ-phenylenediamine (dPhenyle): 7.81 μg/mL; ρ-phenylenediamine: 2.9 μg/mL; double resorcinol(dresorcinol): 62.5 μg/mL; resorcinol: 51.25 μg/mL; double 2-mercaptobenzothiazole (dMercap): 3.90 μg/mL; 2-mercaptobenzothiazole: 1.95 μg/mL; double diphenylcyclopropenone(dDipheny): 1.95 μg/mL; diphenylcyclopropenone: 0.98 μg/mL; double eugenol: 62.5 μg/mL; eugenol: 31.2 μg/mL; phenylacetaldehyde: 7.8 μg/mL; double isoeugenol: 3.9 μg/mL; isoeugenol: 1.95 μg/mL; 1,2-dibromo-2,4-dicyanobutan(bromoml): 3.9 μg/mL; SDS: 31.25 μg/mL.

### Differential Expression Gene Analysis

RNA-Seq analysis identified 1,224 differentially expressed genes (DEGs) out of 19,267 genes between the allergenic (positive) and non-allergenic (negative) group. ($|\log_2 FC| > 1$, adjusted $p$-value < 0.05). Specific gene expression differences are detailed in the Supplemental Material.

The PCA results depicted in Fig. 3A illustrated the distribution of chemical compounds from the positive and negative groups. The first two principal components (PC1 and PC2) accounted for 84% of the total variance, with PC1 explaining 58% and PC2 ex-plaining 16% of the variance.

The heatmap in Fig. 3B used DEGs that meet the criteria of $|\log_2 FoldChange| > 1.5$ and $p$adj < 0.05. The gene expression profiles shown in this heatmap were consistent with the results observed in Fig. S1, demonstrating a clear distinction between sensitizers and non-sensitizers. Weak sensitizers showed an intermediate expression pattern that was distinguishable from non-sensitizers but less pronounced than that of strong sensitizers. A volcano plot illustrating the distribution of DEGs was shown in Fig. 3C. Red and blue dots represented significantly up-regulated and down-regulated genes respectively, while grey dots indicate genes that did not meet the significance.

### KEGG and GO

As shown in the KEGG enrichment dotplot in Fig. 3D, the DEGs revealed significant enrichment in several pathways (adjusted $p$-value < 0.05). The top enriched pathways included cytokine-cytokine receptor interaction, MAPK signaling pathway, PI3K-Akt signaling pathway, transcriptional misregulation and IL-17 signaling pathway.

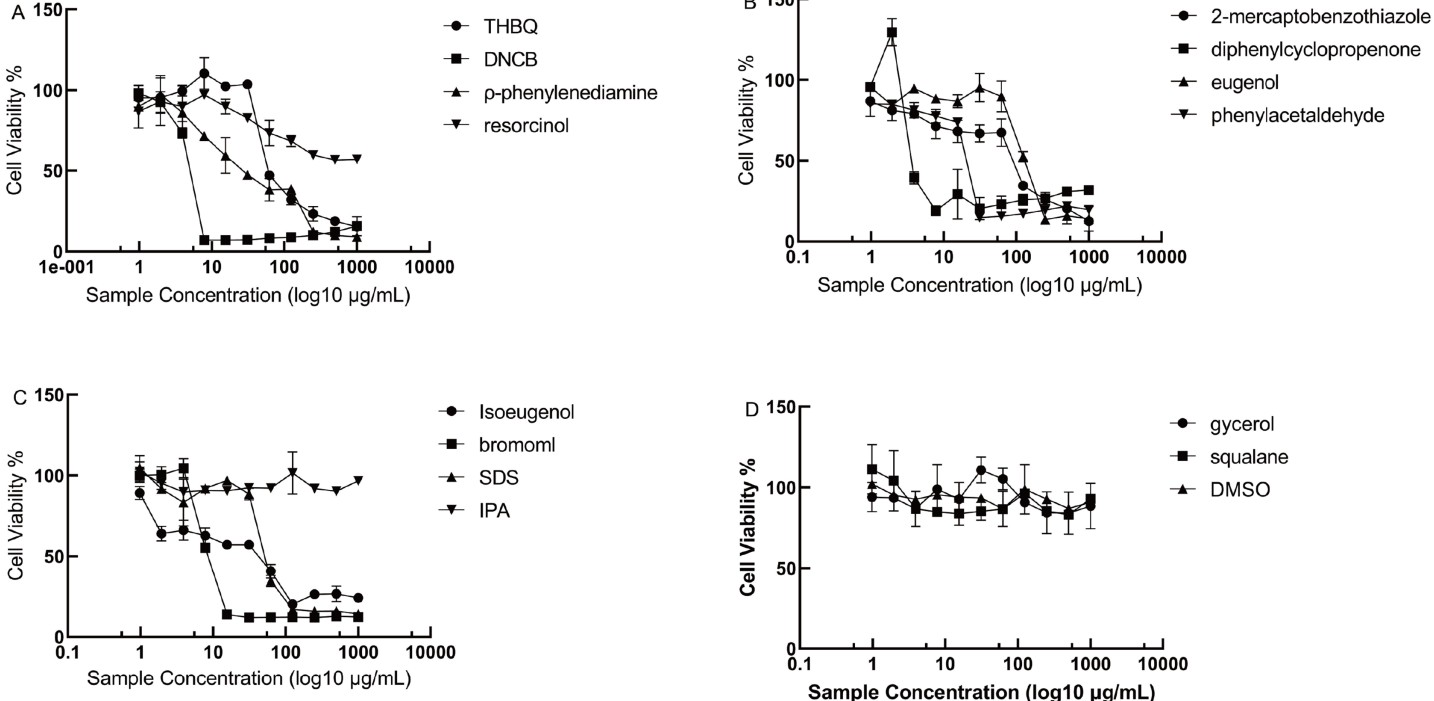

**Figure 2 Dose-response curves of chemical compounds on THP-1 viability.** (A) Cell viability of THBQ, DNCB, ρ-phenylenediamine and resorcinol. (B) Cell Viability of 2-mercaptobenzothiazole, diphenylcyclopropenone, eugenol and phenylacetaldehyde. (C) Cell viability of isoeugenol, bromoml, SDS and IPA. (D) Cell viability of glycerol, squalane, IPA and DMSO. Data points represent mean ± SD from three independent experiments. THBQ, tert-butylhydroquinone; DNCB, 2,4-dinitrochlorobenzene; bromoml, 1,2-dibromo-2,4-dicyanobutane; SDS, sodium do-decyl sulfate; IPA, isopropanol; DMSO, dimethyl sulfoxide.

Significantly over-represented biological processes (BP), molecular functions (MF) and cellular components (CC) were identified by GO enrichment analysis (Fig. 3E). Top enriched GO terms in BP included myeloid leukocyte migration, cell chemotaxis, granulocyte migration and leukocyte chemotaxis with associated genes such as CSF1, SLAMF8, S100A14 and S100A9. Top enriched GO terms in CC were growth factor binding, protein tyrosine, receptor ligand activity and cytokine receptor binding with related gene such as CSF1, DUSP2, NGF and LEFTY1. The details of the cellular components (CC) are shown in Fig. 3E.

## WGCNA and allergic significant module identification

Based on the results from the Figs. 4 and S2, diphenylcyclopropenone (Diphenyl), Eugenol, phenylacetaldehyde (Phenylac), Isoeugenol, and 2,4-dinitrochlorobenzene (Bromoml) were selected as allergenic group, while the remaining negative samples were categorized as non-allergenic group. A total of 28 samples were subsequently used for WGCNA analysis and 26,225 genes were clustered and grouped in-to 16 modules based on their expression patterns (Fig. 4A). The sample dendrogram and heat map of the traits are shown in the Fig. S2. The RsquaredCut was set as 0.83, β = 8 was determined to be the powerEstimate soft-thresholding power (Fig. S3). The correlations between MEs and clinical traits (allergy) were analyzed (Fig. 4B). With respect to the allergy, two modules showed strong

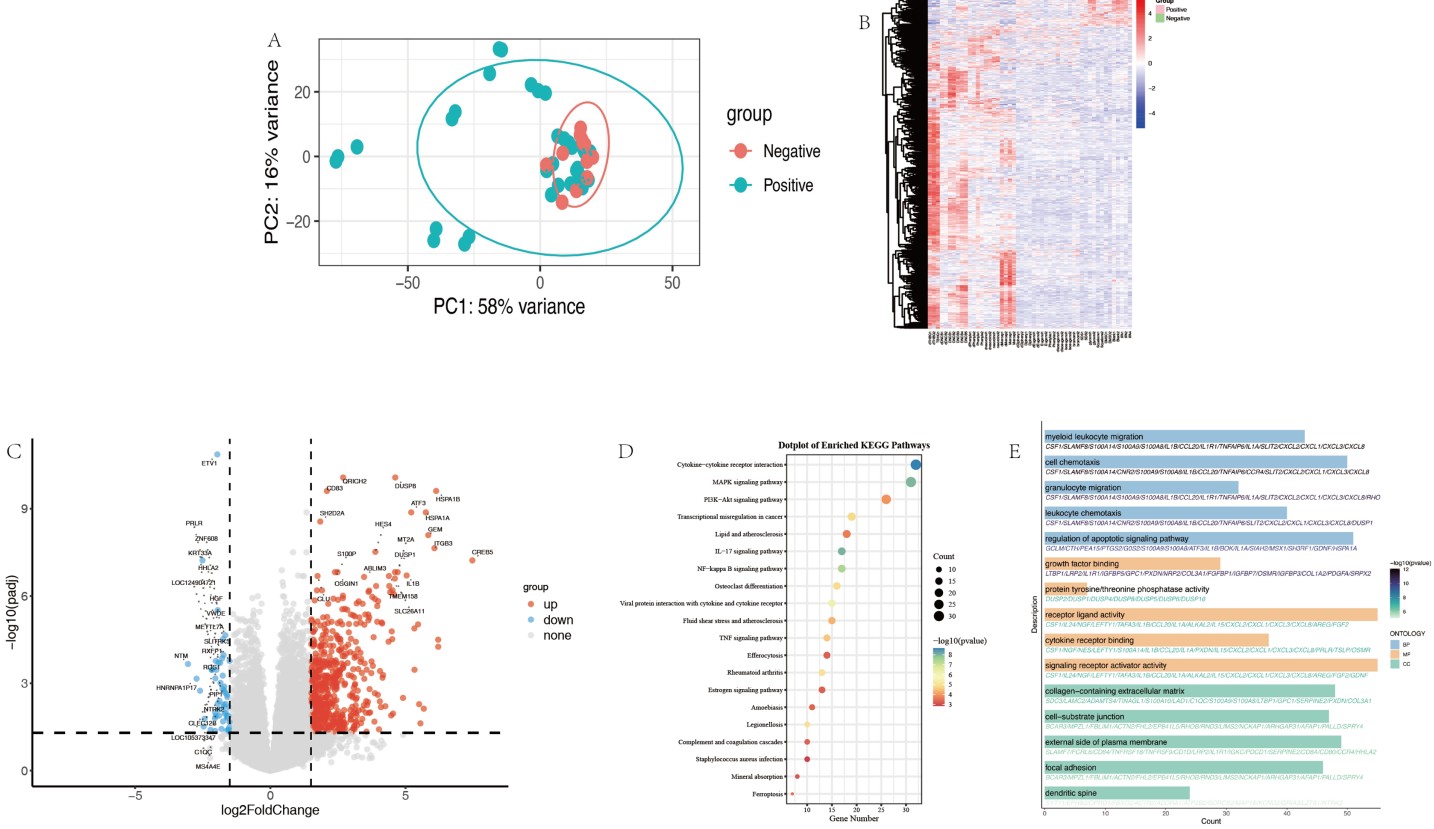

**Figure 3** **Differential expression gene analysis.** (A) The PCA of allergenic (Positive) and non-allergenic (Negative) group. (B) Heatmap of DEGs. (C) The volcano plot of DEGs. (D) Dotplot of enriched KEGG pathways. (E) GO enrichment analysis of DEGs. DEGs, differential expression genes; PCA, principal components analysis.

correlations. The green module showed a significant positive correlation (r = 0.63, $p = 4 \times 10^{-4}$), while the turquoise module showed a strong negative correlation (r = −0.63, $p = 4 \times 10^{-4}$) (Fig. 4B). Compared to the other modules, the green module was associated with allergy and modules showed a high level of positive correlation (Fig. 4C). To focus on the most significant gene-gene interactions, we filtered the edges based on their weights, selecting only those with a weight greater than 0.3. The filtered network was visualized using Cytoscape, providing a clear graphical representation of the gene interactions (Fig. 4D). The edges were sorted in descending order based on their weights and we selected the top 100 genes that were prepared as in-put features for subsequent machine learning analysis.

## Machine learning

As shown in Table 2, we evaluated four machine learning methods (RF, KNN, SVM and NSC/voomNSC) using different feature selection approaches: top 100 variables by variance (Var 100) genes, DEG (100) genes and WGCNA (100) genes. After processing the raw count data with the deseq-vst method, the RF algorithm, trained using WGCNA feature selection method, achieved training accuracy of 86.7% and testing accuracy of 100%. The tmm-logcpm data processing method combined with the DEG feature selection produced

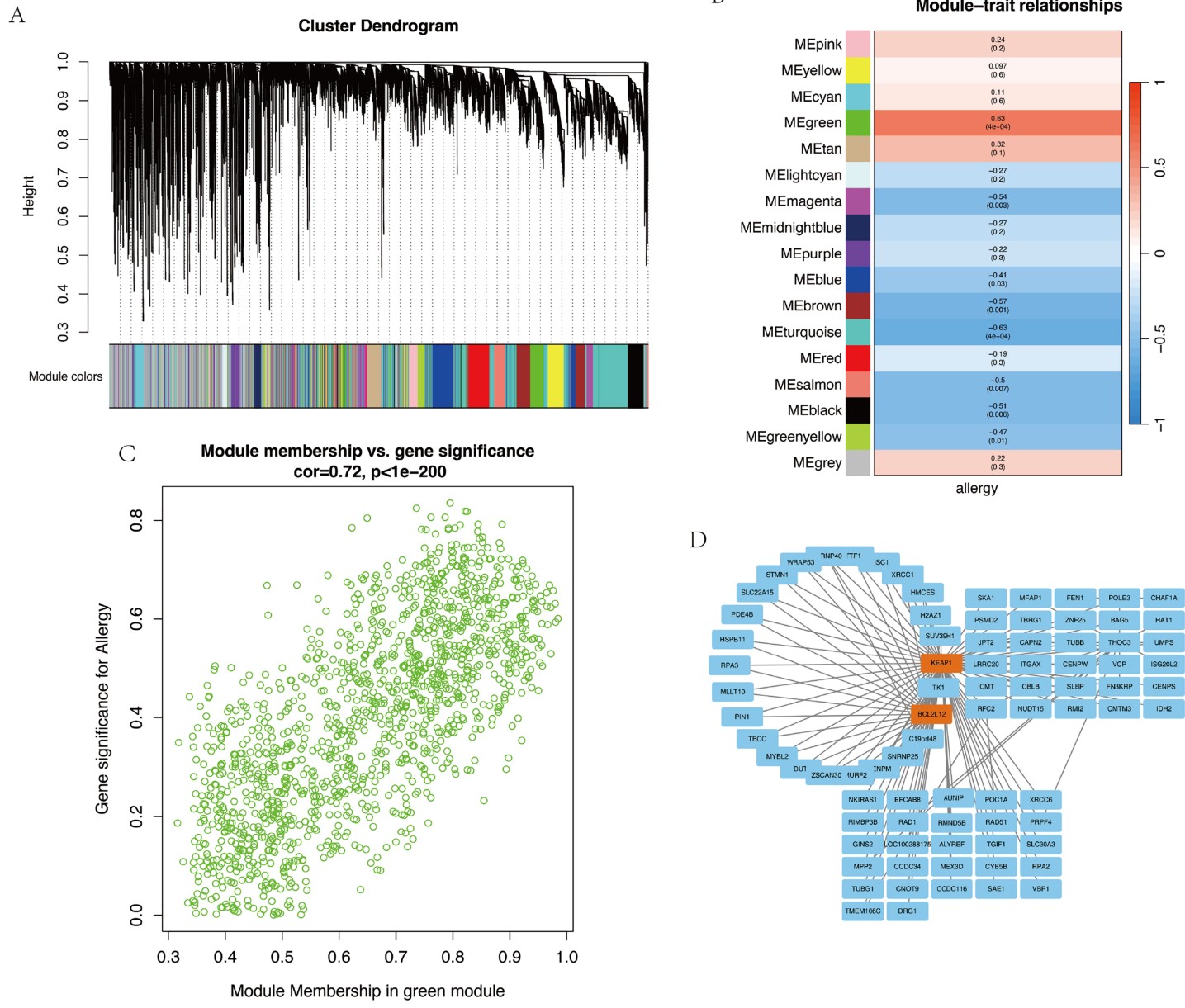

**Figure 4** **The analysis of WGCNA.** (A) Gene dendrogram obtained by clustering the dissimilarity with the corresponding module colors indicated by the color row. (B) Relationships of consensus module eigengenes and allergy. (C) The scatter plot of module membership in green module and gene significance for allergy. (D) Protein-protein interaction (PPI) network of genes in the green. WGCNA, weighted correlation network analysis.

the best results (training accuracy: 83.3%; test accuracy: 95.2%), as shown in Table 3. Using deseq-vst with DEG features selection, KNN achieved a test accuracy of only 61.9%. In contrast, using the tmm-logcpm processing method, the KNN model achieved a test accuracy of 81%.

The SVM classifier exhibited excellent performance across different kernel functions, feature extraction methods, and data processing methods. With the exception of the combination of deseq-vst data processing and Vars feature extraction method, all other

**Table 2 Comparison of training and test accuracies for three feature selection methods using deseq-vst data processing.**

| ML classifier (deseq-vst) | 1st Method with Vars (100) genes | | 2nd Method with DEG (100) genes | | 3rd Method with WGCNA (100) genes | |
|---|---|---|---|---|---|---|
| | Train accuracy | Test accuracy | Train accuracy | Test accuracy | Train accuracy | Test accuracy |
| Random forest (RF) | 0.871 | 0.810 | 0.867 | 0.857 | 0.862 | 0.905 |
| K-nearest neighbors (KNN) | 0.867 | 0.810 | 0.867 | 0.857 | 0.867 | 0.857 |
| Support vector machine (SVM)(svmRadial/Poly) | 0.871/0.933 | 0.905/0.857 | 0.931/0.933 | 0.905/0.905 | 0.90/0.90 | 0.905/0.905 |
| Nearest shrunken centroids (NSC)/voomNSC | 0.833/0.90 | 0.619/0.9524 | 0.90/0.90 | 0.905/0.619 | 0.833/0.833 | 0.619/0.619 |

**Table 3 Comparison of training and test accuracies for three feature selection methods using tmm-logcpm data processing.**

| ML classifier (tmm-logcpm) | 1st Method with Vars (100) genes | | 2nd Method with DEG (100) genes | | 3rd Method with WGCNA (100) genes | |
|---|---|---|---|---|---|---|
| | Train accuracy | Test accuracy | Train accuracy | Test accuracy | Train accuracy | Test accuracy |
| Random forest (RF) | 0.833 | 0.905 | 0.867 | 0.952 | 0.862 | 0.905 |
| K-nearest neighbors (KNN) | 0.833 | 0.667 | 0.867 | 0.810 | 0.867 | 0.857 |
| Support vector machine (SVM)(svmRadial/Poly) | 0.867/0.90 | 0.905/0.952 | 0.90/0.90 | 0.905/0.905 | 0.896/0.866 | 0.952/0.857 |
| Nearest shrunken centroids (NSC)/voomNSC | 0.867/0.8333 | 0.952/0.619 | 0.867/0.967 | 0.905/0.810 | 0.833/0.833 | 0.619/0.619 |

feature extraction techniques resulted in test accuracy exceeding 90% for both kernel types (Tables 2 and 3). This study used a variety of linear discriminant analysis (LDA) algorithms, including negative binomial linear discriminant analysis (NBLDA), Poisson linear discriminant analysis (PLLDA), voomDLDA and voomDQDA, to classify the allergenic compounds. The combination of deseq data processing method and DEG feature extraction method caused a notable improvement for PLDA, achieving a 90.5% test accuracy. Notably, the voomDQDA model demonstrated superior performance when using deseq-vst processed data, achieving test and training accuracy of over 90% (Tables 4 and 5). Table S1 presented that the use of the Var feature selection method, in which 50 genes were selected as features and processed with DESeq-vst, achieving high accuracy. Table S2, employing 60% of the data as test data, showed that the DEG (100) model achieved an accuracy exceeding 85%, while the Var (100) model attained an accuracy above 75%.

# DISCUSSION

In this study, eight machine learning models were utilized to predict the allergenic potential of chemical compounds. Predicting the potential for skin sensitization is crucial for the risk assessment of the use of these compounds. The RNA-seq data utilized in this study were obtained from THP-1 cells derived from HaCaT/THP-1 co-culture system. In recent years, RNA-Seq technology has matured significantly and has become a routine method for detecting gene expression changes (*Wang, Gerstein & Snyder, 2009*). We utilized RNA-Seq data as input for our machine learning models to explore new perspectives that could be used in allergy prediction.

**Table 4 Comparison of training and test accuracies for three feature selection methods using deseq data processing.**

| ML classifier (deseq) | 1st Method with Vars(100) genes | | 2nd Method with DEG(100) genes | | 3rd Method with WGCNA(100) genes | |
|---|---|---|---|---|---|---|
| | Train accuracy | Test accuracy | Train accuracy | Test accuracy | Train accuracy | Test accuracy |
| Poisson linear discriminant analysis (PLDA) | 0.733 | 0.810 | 0.6 | 0.571 | 0.633 | 0.857 |
| Negative binomial linear discriminant analysis (NBLDA) | 0.621 | 0.762 | 0.516 | 0.905 | 0.700 | 0.810 |
| Voom-based diagonal linear discriminant analysis (voomDLDA) | 0.700 | 0.810 | 0.867 | 0.905 | 0.700 | 0.810 |
| Voom based diagonal quadratic discriminant analysis (voomDQDA) | 0.800 | 0.905 | 0.837 | 1 | 0.833 | 0.9524 |

Among common sensitizers, pre-haptens and pro-haptens (Table 1) require spontaneous air oxidation or enzyme-mediated activation to convert into sensitizers, as keratinocytes cells containing key metabolic enzymes such as cytochrome P450 (*Eskes et al., 2019*; *Swanson, 2004*). In the co-culture system involving THP-1 and HaCaT/keratinocytes cells (*Hennen & Blömeke, 2017*), which represent KE2 and KE3 in AOP for skin sensitization. Eskes et al. demonstrated that HaCaT cells co-cultured with THP-1 cells have the potential to increase the response to pro-haptens chemicals (*Eskes et al., 2019*). In contrast, *Sawada et al. (2022)* utilized keratinized normal human epidermal keratinocytes (NHEK) and THP-1 cells. We linked the AKR1C2 promoter to luciferase plasmid, electroporated plasmid into HaCaT cells, and selected the cells, which were used to represent KE2. The HaCaT cells can also be used in OECD 442D luciferase assays to assist in verifying skin sensitization. Most skin sensitization assays involving THP-1 co-culture systems use flow cytometry to detect CD86 and CD54, which rely on single or few markers for classification (*Nukada et al., 2011*). Therefore, we used RNA-Seq for comprehensive analysis of co-cultured THP-1 transcript expression, which can help identify better biomarkers and provide deeper insights into the underlying mechanisms. Simultaneously, to broaden the model's applicability and mitigate bias due to the limited sample size, we conducted experience on a more diverse selection of chemical compounds including varied molecular structures, antigenic properties, and potency categories.

Microarrays based on hybridization of fluorescently labeled cDNA to pre-designed probes on a chip have been widely used for clustering and classification of continuous expression data (*Stears, Martinsky & Schena, 2003*). The initial GARD workflow used Affymetrix's RMA algorithm for normalization (*Johansson et al., 2013*). Subsequently, the advanced GARD workflow converted microarray data to the NanoString nCounter platform to facilitate better assessment of biomarker characteristics (*Forreryd et al., 2016*). *Robinson, Wang & Storey (2015)* discovered that PCR quantitative validation of microarrays may exhibit greater systematic bias than RNA-seq, particularly for genes with low expression intensity. Due to superior performance, integrative RNA-seq analysis and machine learning methods are applied to explore the key genes, disease classification (*Bostanci et al., 2023*), and potential mechanisms.

**Table 5 Comparison of training and test accuracies for three feature selection methods using tmm data processing.**

| ML classifier (tmm) | 1st Method with Vars (100) genes | | 2nd Method with DEG (100) genes | | 3rd Method with WGCNA (100) genes | |
|---|---|---|---|---|---|---|
| | Train accuracy | Test accuracy | Train accuracy | Test accuracy | Train accuracy | Test accuracy |
| Poisson linear discriminant analysis (PLDA) | 0.767 | 0.619 | 0.6 | 0.667 | 0.645 | 0.5238 |
| Negative binomial linear discriminant analysis (NBLDA) | 0.567 | 0.524 | 0.567 | 0.714 | 0.600 | 0.5238 |
| Voom-based diagonal linear discriminant analysis (voomDLDA) | 0.667 | 0.810 | 0.867 | 0.905 | 0.710 | 0.810 |
| Voom based diagonal quadratic discriminant analysis (voomDQDA) | 0.806 | 0.857 | 0.833 | 1 | 0.833 | 0.905 |

In the context of machine learning analysis of RNA-Seq data, feature extraction is a critical step. When selecting feature extraction methods, factors such as dataset size and complexity, computational efficiency, and diversity of results are important concerns. Methods such as recursive feature elimination for gene selection (*Escanilla et al., 2018*) and PCA for dimensions reduction to extract important features are viable options. In this study, we compared three methods: the most basic variance (var100), DE analysis, and WGCNA. Var is the most direct and simplest method, focusing purely on the data perspective. DE analysis is a conventional approach for group comparisons. Figure 3A shows the PCA distribution where the samples are not completely separated from the perspective of PC1 and PC2, which is generally consistent with the result of Henrik. Figures 3B and 3C presented heatmap and volcano plot related to the distribution of differentially expressed genes. Figures 3D and 3E show the enrichment of differential genes in pathways such as cell chemotaxis, granulocyte migration, and MAPK signaling pathway, which are all related to DC maturation and allergy (*Miyazawa et al., 2007*).

Many algorithms have been proposed to classify based on continuous log intensities obtained from microarray experiments. It is reported that RNA sequencing data, derived from genome alignment raw counts, are excessively dispersed, with the count variance exceeding the mean (*Goksuluk et al., 2019*). Therefore, using appropriate data processing can not only enhance accuracy but also enable analysis in conjunction with microarray algorithms. In this study, we applied machine learning algorithms to the classification of RNA-Seq data using the MLSeq (2.22.0) R package. Normalization methods containing deseq median ratio (deseq) and trimmed mean of M means (TMM) were used to correct systematic variations biases in RNA-Seq data. After completing the normalization process, NBLDA and PLDA can be used directly for classification. Witten assumed that RNA-Seq counts follow a Poisson distribution and consequently proposed sparse PLDA as a classification algorithm for RNA-Seq data (*Witten, 2011*). However, PLDA performs deteriorates when the counts are over-dispersed. The results from Tables 1 and 2 show that among the three feature extraction methods, only WGCNA achieved 90% performance with PLDA, while the others performed relatively poorly. Dong used edgeR to perform the gene selection for NBLDA classifier and He also noted the use of network information of

pathways for better testing. In our results, we found that the WGCNA feature processing method combined with the DESeq normalization method achieved relatively good results (NBLDA).

Common transformation techniques included variance stabilizing transformation (vst) (differential expression), logarithm of counts per million reads (logcpm) (*Robinson, McCarthy & Smyth, 2010*) and variance modelling at observational level (voom) (*Law et al., 2014*) that can be used to make discrete RNA-seq data hierarchically more similar to microarray data. VoomNSC is a sparse classifier that models the relationship between the mean and the variance using the voom method. It incorporates the precision weights from voom into the NSC classifier through weighted statistics (voomDDA). Dudoit extended diagonal discriminant classifiers to voomDLDA and voomDQDA, which use all features for classification (*Dudoit, Fridlyand & Speed, 2002*). In the research results, voomDQDA and voomDLDA demonstrated better training performance compared to voomNSC, regardless of the feature selection method or data processing approach used. Notably, voomDQDA consistently achieved accuracy rates above 85%. We believe this advantage stems from two main factors: the robustness of the voom method and the strengths of diagonal algorithms. Traditional machine learning methods such as KNN, RF, and SVM were also used for testing. Both GARD and the improved GARD utilized SVM as the classifier, demonstrating SVM excellent performance. The effect of different kernels (Radial and Poly) on test and train accuracy was also compared. The results showed that the SVM accuracy was below 90% when using the vars feature selection combined with deseq-vst data processing. Additionally, the Poly kernel required a longer model construction time. In *Forreryd*'s *(2016)* study, the RF model achieved an accuracy of 100%. However, in our study, RF reached 100% accuracy only when using deseq-vst combined with WGCNA for feature selection. Balanced accuracy (Table S3) can balance sensitivity and specificity, providing a more equitable view of model performance in the presence of class imbalance. Although we have employed various models and rigorous validation techniques to mitigate the limitations posed by the sample size and to ensure that the model can capture relevant patterns, further testing of more molecules is still needed to ensure broader ranges.

## CONCLUSIONS

Our research presents an integrated approach combining co-culture systems, RNA-Seq, and machine learning techniques to provide a more comprehensive assessment of allergenic compounds. This multifaceted strategy offers a holistic perspective, high-throughput data, and advanced analysis capabilities. However, we acknowledge several challenges, including sample size limitations, model accuracy, and model interpretability. To date, there has been no alternative testing method that can sub-categorize skin sensitizers into UN GHS subcategories 1A and 1B. By pursuing these future directions, we aim to enhance the robustness, accuracy, and applicability of our approach, ultimately contributing to safer consumer products and more effective risk assessment strategies.

### Funding

The authors received no funding for this work

### Competing Interests

The authors declare that they have no competing interests. The authors are all employed by Pigeon Manufacturing (Shanghai) Co., Ltd.

### Author Contributions

- Wu Qiao conceived and designed the experiments, performed the experiments, analyzed the data, prepared figures and/or tables, authored or reviewed drafts of the article, and approved the final draft.
- Tong Xie analyzed the data, prepared figures and/or tables, and approved the final draft.
- Jing Lu performed the experiments, authored or reviewed drafts of the article, and approved the final draft.
- Tinghan Jia conceived and designed the experiments, authored or reviewed drafts of the article, and approved the final draft.

### Data Availability

The raw sequence reads are available at NCBI: PRJNA1148804; SRR30271542–SRR30271594.

https://www.ncbi.nlm.nih.gov/sra/?term=PRJNA1148804.

The raw data are available in the Supplemental Files.

### Supplemental Information

Supplemental information for this article can be found online at http://dx.doi.org/10.7717/peerj.18672#supplemental-information.

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
