# Peer review of "Development of machine learning models for the prediction of the skin sensitization potential of cosmetic compounds"

_PeerJ, doi:10.7717/peerj.18672_

## Round 0.1 · original submission · Major Revisions

Besides all comments rasied by reviewers, please explain in more detail on two issues:

1. use of small sample size may impact the generalization of findings to broader spaces.
2. For sensitivity analysis, try to use 50% or less as training data and rest as testing data.

Reviewer 1 ·

Basic reporting

This study developed a machine-learning model to assess skin sensitization using RNA-Seq expression data as a feature. There are several concerns regarding the study design that need to be addressed.

Experimental design

The choice to utilize only 16 molecules to construct ML models raises questions. It is unclear whether these molecules vary significantly in their chemical properties, such as physicochemical properties. Clarification is needed on whether these properties are similar or diverse among the selected molecules. This information is crucial as it affects the model's ability to generalize. If the prediction molecules fall within the same range of physicochemical properties or share molecular substructures, they may exhibit similar skin sensitization effects, which could be biased and limit the applicability domain of this model.

The model was built using only 60% of the compounds, equivalent to 10 molecules. This is a relatively small sample size, which makes me doubt the model’s generalizability across broader chemical spaces.

According to the OECD guidelines on in vitro testing of skin sensitization, the human cell line activation test (h-CLAT) or the local lymph node assay (LLNA) are commonly recommended for skin sensitization assessment. It is important to discuss what advantages, if any, the co-culture of THP-1 and HaCaT cells offers over these traditional methods. Additionally, the relevance of cell viability as an endpoint for skin allergy needs to be reconsidered and discussed, as it is not typically used for skin allergy assessments.

Validity of the findings

The model incorporates a high number of features (100 features) relative to a very small training set (10 molecules). Despite this, the model shows comparable evaluation scores between the training and test sets (Table 2). This outcome should be discussed in the manuscript to explain how such consistency is achieved despite the apparent limitations.

Some chemicals displayed very high cytotoxicity (Figure 2), which could alter the gene expression of the cells. How did the author discriminate the effects of this phenomenon from those of skin sensitization or skin allergy? If the chemicals exhibit such high cytotoxicity, are they considered to cause skin corrosion or irritation instead of skin sensitization?

Additional comments

An accuracy metric does not cover an imbalanced dataset, where the positive and negative are very different (10 v 5). The author should consider balanced accuracy or sensitivity and specificity, instead of relying on the accuracy metric alone.

·

Basic reporting

Yes, the manuscript is clear and unambiguous, background and information provided adequate for understanding the study, . Figures and tables are clear and legible. The current condition is sufficient

Experimental design

Experimental design well defined, relevant and meaningful.

Validity of the findings

the findings described by the author correlate with the result. All underlying data have provided, conclusion align with align findings obtained in the study.

Additional comments

This study provides a new breakthrough in the use of artificial intelligence in clinical research by reducing the use of experimental animals in clinical trials, related to the awareness of researchers in various parts of the world regarding animal welfare. So that the development of the latest technology and the accumulation of toxicity and efficacy test data make it possible to predict the toxicity of target substances without using experimental animals.

1. However, it is necessary for authors to explain the consideration of using 60% data for the training data set while 40% is for the testing data set.
2. Authors are also expected to provide an explanation of the limitations found in this study, what is the ideal sample size, what kind of accurate interpretation is expected and also whether there are problems in determining pre-hapten or hapten in various cosmetic ingredients that were tested in this study.

---

## Round 0.2 · accepted · Accept

The reviewers are satisfied with current version.

Reviewer 1 ·

Basic reporting

After carefully reviewing the revised manuscript, the authors adequately addressed and corrected my concerns. As a result, the manuscript has been significantly strengthened. Therefore, I recommend publishing it in its current form.

Experimental design

The experimental designs are well-constructed and adhere to the standard OECD guidelines for toxicity testing. The machine learning protocol is also acceptable.

Validity of the findings

As the authors have addressed the comments, the findings may be accepted in their current form.

·

Basic reporting

The current condition is sufficient.

Experimental design

-The current condition is sufficient.

Validity of the findings

-The current condition is sufficient.

Additional comments

-